# The Failure Mechanism of Composite Stiffener Components Reinforced with 3D Woven Fabrics

**DOI:** 10.3390/ma12142221

**Published:** 2019-07-10

**Authors:** Qiaole Hu, Hafeezullah Memon, Yiping Qiu, Yi Wei

**Affiliations:** 1Key Laboratory of Textile Science & Technology, Ministry of Education, College of Textiles, Donghua University, 2999 North Renmin Road, Shanghai 201620, China; 2Center for Civil Aviation Composites, Donghua University, 2999 North Renmin Road, Shanghai 201620, China; 3College of Textiles and Apparel, Quanzhou Normal University, Quanzhou, Fujiang 362000, China

**Keywords:** 2.5D woven fabrics, 3D composite, L-beam, transverse load, interlaminar shear, bending, stiffeners

## Abstract

Composite industry has long been seeking practical solutions to boost laminate through-thickness strengths and interlaminar shear strengths (ILSS), so that composite primary structures, such as stiffeners, can bear higher complex loadings and be more delamination resistant. Three dimensional (3D) woven fabrics were normally employed to render higher transverse and shear strengths, but the difficulty and high expense in producing such fabrics make it a hard choice. Based on a novel idea that the warp yarns that interlock layers of the weft yarns might provide adequate fiber crimps that would allow the interlaminar shear or radial stresses to be transferred and borne by the fibers, rather than by the relatively weaker matrix resin, thus improving the transverse strengths, this work provided a two point five dimensional (2.5D) approach as a practical solution, and demonstrated the superior transverse performances of an economical 2.5D shallow-bend woven fabric (2.5DSBW) epoxy composites, over the conventional two dimensional (2D) laminates and the costly 3D counterpart composites. This approach also produced a potential candidate to fabricate high performance stiffeners, as shown by the test results of L-beams which are common structural components of any stiffeners. This study also discovered that an alternative structure, namely a 2.5D shallow-straight woven fabric (2.5DSSW), did not show any advantages over the two control structures, which were a 2D plain weave (2DPW) and a 3D orthogonal woven fabric (3DOW) made out of the same carbon fibers. Composites of these structures in this study were conveniently fabricated using a vacuum-assisted resin infusion process (VARI). The L-beams were tested using a custom-made test fixture. The strain distribution and failure mode analysis of these beams were conducted using Digital Image Correlation (DIC) and X-ray Computed Tomography Scanning (CT). The results demonstrated that the structures containing Z-yarns or having high yarn crimps or waviness, such as in cases of 3DOW and 2.5DSBW, respectively, were shown to withstand high loadings and to resist delamination, favorable for the applications of high-performance structural composites.

## 1. Introduction

Composite materials play an important role in aerospace, rail, automotive, and energy applications because of their high specific strength, specific modulus, processability, and chemical stability [1,2,3,4,5,6]. Composite stiffeners, such as C, L, π and I beams, are one of the most common and essential structural components in modern aircrafts, rail cars, automobiles and wind blades. Stiffeners often bear complex loadings, both in-plane and out-plane, making them critical parts in primary structure assemblies, such as the spars and ribs in aircraft wings, tails, and fuselages. Currently most of such composite stiffeners are made of unidirectional or 2D woven fabrics, or the combination of both. The main drawbacks of these stiffeners are their delamination at low loadings and low transverse strengths, limiting their uses where shear and transverse loads, such as in bending, are substantial [7]. This disadvantage is especially prominent for composite structures having curved shapes, such as ribs, angle bracket, stiffeners, and wind blades [7]. When these structures are subjected to tensile, compressive bending in the plane of curvature, the interlaminar shear and radial stress develop in the through-thickness direction, namely the Z-direction, especially in the regions where the high bending moments are present, thereby resulting in premature delamination failure.

By comparison, three dimensional (3D) woven fabric composites with fibers in the Z-direction render high through-thickness strengths, damage resistance, delamination resistance, and impact resistance [8,9,10,11,12,13]. These improvements are attributed to the presence of continuous Z-direction fibers, and the 3D composite structure is considered to be the smoking gun for high transverse strengths. Unfortunately, these 3D woven fabrics, such as 3D orthogonal woven (3DOW) fabrics, are very difficult and expensive to produce, particularly when fabric thickness is higher than ten millimeters is needed, making them nearly impossible to be economically manufactured and extensively used in common structures such as stiffeners.

Unidirectional fibers, either in forms of fiber tape or non-crimp fabrics, together with biaxial woven fabrics, make most of today’s composite stiffeners, due to their availability and economical scales. Biaxial fabrics usually render higher resistance than unidirectional fibers because of their yarn-crimps or yarn-waviness towards the Z-direction, providing a certain degree of transverse load bearing. Based on this understanding, it was hypothesized that if adequate yarn waviness were created, namely by weaving a 2.5D fabric of certain structures, the resulting 2.5D composites might be able to provide interlaminar shear strengths and transverse strengths close or equivalent to that of the 3D composites. Since the 2.5D fabrics do not require Z-yarns, they can be produced on the most common weaving looms with minor modifications, thus avoid the technical difficulty and high costs in 3D fabric production, consequently providing a practical and economical alternative solution to the composite stiffener community.

In order to test this theory, a L-shape beam (L-beam) was selected as a model composite configuration. L-beams are essential components in any composite stiffeners, such as C, L, π, and I stiffeners. Their simple geometry allows the design of custom fixtures to generate tensile or compressive bending, so as to test their transverse properties and observe the corresponding failure modes. To our knowledge, there were no previous studies in the literature on the bending of 2.5D woven fabric composites.

Literature [14,15,16,17,18] studied 3D composite L-beams and T-joints under tensile load, demonstrating their ability to carry a significantly higher load until failure than that of the 2D laminate counterparts. They further pointed out that the failure mode in 2D laminate L-beams was dominated by delamination due to out-plane tensile stress. These studies provided insights into the failure mechanisms of composite stiffeners, but the behaviors of the stiffeners under compression, which is the most common load, remained unknown. In the tensile mode, compression is generated by an “outward” bending moment, i.e., while the moment-induced compressive-stress is generated at the outer-surface (at R = Ro) and the tensile-stress generated at the inner-surface (at R = Ri).

Few studies were on the failure mechanisms of structures subjected to an “inward” bending moment, in which cases tension is created at the outer-surface and compression at the inner-surface. Springer and Chang [19] showed, using the finite element method, that under “inward” bending, the beam failed primarily via delamination at small radius-to-thickness ratios (R_i_/H < 0.3–0.5) and the in-plane failure was at large radius-to-thickness ratios. Helenon et al. [20] used experimental and numerical methods to study the failure of 2D composite T-stiffeners at three out-plane bending angles (i.e., β = 0°, 45°, 90°). Their results revealed that the maximum free-edge principal transverse stresses occurred at the failure locations perpendicular to the fiber direction. A conference paper reported 2D L-beams with three different layups were subjected to a compressive load [21], and using cohesive zone modeling, their results revealed that the maximum interlaminar shear strength occurred at the lower end of the laminate and the initial failures for all three layups were dominated by the interlaminar tensile stress. Burns et al. used three novel designs to increase the failure stress and damage limit of 2D T-joints, using bending tests and finite element modeling [22]. Their results showed that the initial damage load was ~125% higher than that of a conventional T-joints. Furthermore, a similar progressive failure mode was observed in all T-joints, in which delamination initiated in the radius-bending region, followed by radius bend/delta-fillet interface cracking. There is an apparent lack of studies on 2.5D and 3D composite curved structures when they are subjected to compressive bending.

This study tried to provide an original and preliminary investigation of 2.5D woven fabric composites’ mechanical performances under compressive bending, along with the 2D plain weave (2DPW) and 3D orthogonal woven (3DOW) composites as references. Two 2.5D woven structures were selected and their fabrics produced for this work, namely a shallow-straight (3DSSW and a shallow-bend (3DSBW). The effect of woven structures on the strain distribution and failure modes was evaluated via Digital Image Correlation (DIC) and X-ray Computed Tomography (CT) scans. Considerable efforts were made to produce the carbon fiber woven fabrics and the epoxy composites so that their fiber volume could be controlled at the same level, therefore rendering meaningful and valid comparison of their properties.

## 2. Materials and Experimental Methods

### 2.1. Materials

The fibers used in this work were SYT55S-12K carbon fibers made by Zhongfu Shenying Carbon Fiber Co., Ltd. (Lianyungang, China), and the matrix resin is BAC 172, an infusible one-component epoxy resin supplied by Zhejiang Baihe Advanced Composites Ltd. (Zhejiang, China) The properties of the raw materials are listed in Table 1 and Table 2.

### 2.2. Fabric Preparation

The four woven structures, 2DPW, 2.5DSSW, 2.5DSBW, and 3DOW (see Figure 1a–d), were fabricated along the warp/X-direction, into the L-shaped fabric preforms. The 2DPW fabrics were constructed by interlacing two types of yarns, i.e., warp yarns (in yellow color) and the weft yarns (in blue color), repetitively over and under each other. The 2.5DSSW and 2.5DSBW, shown in Figure 1b,c, respectively, also contained two types of yarns, as in the case of the plain weave. The difference was that in these two structures, the warp yarns were placed sinusoidally at certain undulation angles to the thickness direction, i.e., Z-direction, interlacing the adjacent layers of weft yarns. The 2.5DSSW fabric consisted of eight warp layers and eight weft layers, in which two adjacent weft yarns in the Z-direction were held together with undulated warp yarn. It differed from the 2.5DSBW fabric that had eight warp layers and nine weft layers, where a higher undulated warp yarn interlaced two adjacent weft layers in the Z-direction.

The 3DOW fabric was woven with orthogonal fibers consisting of in-plane non-crimp warp and weft yarns interlaced in the through-thickness direction with the binder yarns, i.e., the Z-yarns (in red color; see Figure 1d).

The 2D plain weave fabrics were prepared using a conventional weaving loom (i.e., rapier loom), by Zhongfu Shenying Carbon Fiber Co., Ltd. (Jiangsu, China) The 2.5DSSW, 2.5DSBW fabric preforms were made using jacquard looms, and the 3DOW fabric preforms using a specialty loom, all by Huaheng High-Performance Fiber Textile Co., Ltd. (Yixing, China) The weaving parameters are presented in Table 3. It should be noted that the 2.5D fabrics could be woven on modified conventional rapier looms for even lower costs.

### 2.3. Composite Preparation

A convex heal-resisting forging aluminum alloy mold was designed to mold the L-beams and the composite part was fabricated using the vacuum assisted resin infusion (VARI) technique, in which the fabric preforms were placed in a vacuum bag and infused with resin at room temperature. The vacuum was maintained at 0.098 MPa by a vacuum pump (Welch, MPC301Z, Shanghai, China) during the infusion. The subsequent curing was performed in an oven (Dephyris, DFS180R, Jiangsu, China) at 120 °C for 1 h with the heating rate of 2°/min. Test coupons were cut from the molded part by a water jet cutter (SQ4020, Sunrise Water Jet, Huai’an, China). The final fiber volume fraction in the composites was calculated gravimetrically, with 60% for 2DPW composite (2DPWC) and 54% for the others. The composite preparation was conducted according to the processes described in an earlier publication [8]. Again, it was critical to produce the carbon fiber woven fabrics and the epoxy composites with careful calculation and control of the weaving parameters, as well as the resin infusion process, so that the composite L-beams had fiber volumes at the same level, and meaningful and valid comparison of their properties could be realized.

## 3. Characterization

### 3.1. Compressive Bending Test

The compressive bending test on the L-beams was conducted using a custom-built test fixture, made of 7090 aluminum alloy (see Figure 2a). The loading and holder arm lengths L, L’, width w, thickness t and outer radius R_o_, were listed in Table 4. All test coupon dimensions were given in millimeters and were measured with a Vernier caliper with an accuracy of 0.1 mm.

The quasi-static compressive bending test was performed using a computer-controlled electronic universal testing frame (LD26-5105, Labscans, Shenzhen, China) with a 10 kN load cell. A loader nose with a radius of 5 mm was used in this test and a displacement control of 1 mm/min was employed, as shown in Figure 2b. At least three coupons were tested for each woven type.

### 3.2. Deformation Measurement by Digital Image Correlation (DIC)

Due to the difficulty in directly measuring the L-beam strain distribution, as well as observing the damage, the strain correlation was determined using a non-contact 2D optical deformation measurement software (i.e., GOM correlate) combined with a digital SLR (DSLR) camera (Canon 750D, lens 18–55 mm, ISO 100-250). Considering the camera resolution (24.3 megapixels), a facet size and facet distance of 15 × 15 pixels and 15 pixels, respectively, were employed. In addition, the DIC test coupons needed to have a random gray scale pattern with a pixel size between 3 × 3 pixels and 4 × 4 pixels, in order to be identified by the software according to References [23,24], as shown in Figure 3a. An actual testing coupon for the optical deformation measurement is shown in Figure 3b. All the set parameters for analysis of images was followed by Reference [25], therefore, the re-projection error caused by DSLR cameras was around 0.05 pixels, the displacement and strain difference were around 6% and 0.06%, respectively. More details regarding the level of error in the DIC method when in combination with inexpensive DSLR cameras can be found in the literature [24,25].

### 3.3. Micro X-Ray Computed Tomography (CT) Scan

Three-dimensional images of the cracks and the information on their propagation inside the L-beams after failure were obtained via Micro X-ray CT Scanner (Diondo d2, Hattingen, Germany), a non-destructive X-ray perspective technique. In the X-ray CT test, the spatial resolution was kept at 7 μm. The X-ray CT scan was schematically shown in an earlier publication [8].

## 4. Results and Discussions

### 4.1. Morphology of the L-Beam Cross-Sections

During the compressive bending test, the load was transferred to and borne by the fibers, therefore, the yarn path, namely the fiber buckling, along the Z-direction had an effect on the crack propagation and mechanical properties. Since fiber buckling was largely determined by the weaving and/or braiding structures, and the manufacturing process of composites lead to different internal structures and subsequently different progressive damage processes [26]. Therefore, the understanding of the weaving/braiding topology was essential to determine the correlation between the performance and failure of L-beams under compressive loading.

To observe the yarn pattern in the L-beams, the main factors contributing to the compaction behavior of 2D preforms reported in References [27,28,29,30,31,32] were used as a reference (see Table 5 for the corresponding schematics).

#### 4.1.1. Yarn Path in the L-Beams

The warp yarn path of these four types of beams along A-A’ plane was shown in Figure 4a. Warp yarn (X-direction) paths, characteristic by their sinusoidal curves, were observed in the 2DPW composites (2DPWC), 2.5DSSW composites (2.5DSSWC), and 2.5DSBW composites (2.5DSBWC), whereas straight warp yarns were observed in the 3DOW composites (3DOWC). However, the U-shaped Z-yarn paths in 3DOWC were compressed and tuned into the familiar trapezoid shape, owing to the Z-yarns was squeezed by the adjacent warp yarns and weft yarns during the VARI process. The warp yarn waviness could be ranked in ascending order as follows: 3DOWC < 2DPWC ≤ 2.5DSSWC < 2.5DSBWC. The yarn waviness of 2.5DSSWC was closest to that of 2DPWC, due to the yarn interlacing and yarn compression during the VARI process.

Moreover, the straight weft yarn path along the B-B’ plane occurred in all four composites, as shown in Figure 4b. This might be resulted from the yarn compression during the VARI process, leading to slight reduction in the weft yarn waviness.

#### 4.1.2. The Cross-Sectional Shape of Yarns

The lenticular-shape of cross-sectional weft yarns in 2DPWC, 2.5DSSWC, and 2.5DSBWC (see Figure 4a) were the results of the void/gap accumulation, fiber compression, and the yarn cross-sectional deformation. However, the elliptical cross-sections of weft yarns with various dimensions observed along the Z-direction in 3DOWC, were the results of these yarns being squeezed by the adjacent warp yarns and Z-yarns.

In addition, the cross-sectional shape of warp yarns along the B-B’ plane (Figure 4b) were significantly different from each other for the four structures. For example, the lenticular-shaped cross-sections of the warp yarns, due to the squeezing of the crossed weft yarns, were apparent in 2DPWC. The cross-sections of the warp yarns observed in 2.5DSSWC and 2.5DSBWC had racetrack shapes, owing to compression of these yarns by the parallel weft yarns only. Moreover, the warp yarns in 3DOWC were compressed by the parallel weft yarns and their movement was limited by the Z-yarns, therefore, resulting in the parallelogram-like cross-sectional shape of the warp yarns in this composite.

Furthermore, the cross-sections of both warp and weft yarns were inclined along the Z-direction, and the yarns along the warp/weft direction appeared uneven. This was attributed to the combined effect of bending deformation, yarn nesting, and slipping, all associated with the VARI process.

In conclusion, the morphology analysis of the L-beam cross-sections suggested that the load transfer and failure mode might be significant influenced by the yarn cross-sections and yarn paths under compressive bending and on the failure mode.

### 4.2. Compressive Bending Test

The load-displacement curves of the four L-beams were shown in Figure 5a. The load of the 2.5D and 3D composites increased linearly with displacements of up to 5 mm, and then increased non-linearly up to failure. Contrarily, for the 2DPWC L-beam, the load increased linearly until the test coupon failed, which was attributed to the low yarn waviness and no yarns interlocking between layers.

The compressive bending loads of the four L-beams were presented in Table 6. To illustrate the transverse performance and obtaining a better picture about these structures, the tensile load of these L-beams, which was tested and reported in our earlier publication [8], was also included in this table. As could be seen in Table 6, the 2.5DSBWC and 3DOWC exhibited higher compressive bending load, as well as tensile load, than those of the 2DPWC and 2.5DSSWC. This was attributed to the presence of Z-yarns or high yarn waviness that restrained the crack growth in all directions, therefore, more energy was required for cracks to propagate in 2.5DSBWC and 3DOWC.

Due to the variation in the L-beam thickness, the compressive stress (σ*_c_* = *F*/(*w* × *t*)) was calculated according to the formula used by Springer GS [20] for a more precise comparison. As Table 6 showed, the 2.5DSBWC L-beams unexpectedly had higher compressive stress than that of the 3DOWC. Because there had been no previous theory or study that predicted or showed the effect of load bearing by highly crimped fibers, such as in the case of 2.5DSBWC, could be more significant than the presence of Z-yarns, as in the case of 3DOWC. As confirmed by the equally unexpected high stress form the tensile test of the L-beams, where 2.5DSBWC also exhibited higher tensile stress than that of 3DOW, as shown in Table 6, it might be inevitable to conclude that the high density of the crimped warp fibers towards the Z-direction overpassed the effect due to the lower density of the Z-yarns in 3DOWC.

On the other hand, the compressive stress of 2.5DSSWC was surprisingly low, similar to that of 2DPWC, indicating that the inherently low yarn crimps in 2.5DSSW was further suppressed by the VARI process, reducing the transverse load transfer. Therefore, from the performance and economical point of view, the 2.5DSSW was an inferior structure, and 2.5DSBW fabric would be the choice of reinforcement when it came to the selection of complex load bearing structures, such as stiffeners.

### 4.3. Analysis of Deformation Resistance

The bending angle, θ, calculated from the displacement, was used to estimate the effect of woven structure on deformation resistance. As shown in Figure 5b, two points (i.e., point 1 and point 2) were selected on the loading arm and the corresponding displacement, D1 and D2, were obtained during the test by the GOM software. Therefore, according to the geometric relationship, the bending angle, θ, was calculated by the following equation:(1)$$\theta =\arctan\left(\frac{D2-D1}{D}\right)\times 180/\pi$$
where *D* represented the distance between point 1 and point 2, and was a constant (20 mm); *D*1 and *D*2 were the Y-directional displacement of point 1 and point 2, respectively.

Thus, the bending angle (θ) was calculated. Additionally, the bending angle as a function of load was plotted and shown in Figure 5b. It could easily be seen that the bending angle was affected by the woven structures. In particular, those structures with high yarn waviness or with the presence of Z-yarns had high bending angle (i.e., 2.5DSBWC and 3DOWC). In addition, even with the same bending angle, the load could be different. For example, under the same load, the 2DPWC had the highest bending angle amongst these four composites, indicating that this structure had the lowest resistance to deformation. Therefore, a structure having Z-yarns or high yarn crimp in the thickness direction would exhibit high resistance to deformation.

### 4.4. Digital Image Correlation (DIC) Analysis

Figure 6 were the full field strain maps, which showed the local strain distribution on test coupon surfaces of the four types of L-beams at failure. As Figure 6 showed, the strain concentrations (i.e., red area) differed significantly from each other. For the 2DPWC beam, the maximum strain occurred at the inner-surface of the supporting arm, rather than in the curved area. This indicated that the maximum stress and the failure were primarily generated in this area when this beam was subjected to bending, as shown in Figure 6a.

For 2.5DSSWC and 2.5DSBWC, the maximum strains, different from 2DPWC, occurred at the inner edge of the beams and much closer to the center of the curve, as shown in Figure 6b,c. This behavior obviously was resulted from yarn interlock in the weaving styles, which consequently altered the spots where stress concentrations occurred.

The most significant difference was seen with strain distribution of 3DOWC, in which case the maximum strain appeared on the outer coupon surface, which aligned along the Z-yarn direction. This suggested that the surface strain distributions of the 3DOWC beams were strongly correlated with the constituent Z-yarns, resulting in a stress concentration and failure along the Z-direction.

To elucidate the strain variation of these L-beams under compressive bending, the strains selected at three points of interest on the beam were extracted and plotted (see Figure 7). These selected points were located in the center of the curve area (i.e., points 1–3). It is worth noting that the strain values were calculated from the Mise Stress. Under the compressive bending test, the outer surface was under tension and its strain was labeled as positive, while the inner surface was under compression and its strain was labeled as negative.

For the 2DPWC beam, the strains on the inner-surface (i.e., point-3) and outer-surface (i.e., point-1) increased with increasing loading time (Figure 7a). This indicated that, compared to the other regions, the outer-surface and inner-surface of the beam was subjected to higher stress, either tensile or compressive. Moreover, in terms of strain calculation, the Mise Stress at outer surface was higher than that at the inner surface, indicating that the fiber in the curve area was primarily stretched during the test. However, the strain within the neutral axis area (i.e., point-2) remained approximately constant with increasing loading time, because no stress concentration was generated in this field.

The strain-curves of the 2.5DSSWC beam exhibited a different pattern, as shown in Figure 7b. On the outer surface (i.e., point-1) and inner surface (i.e., point-3), the strains increased slowly between 0–5 min of the loading time, but increased much greater after 5 min. The strain of point-2 is unchanged during the test. These observations indicated that high stress concentration was generated on inner or outer surface, but less stress concentration in neutral axis plane. The strain curves also indicated that 2.5DSSWC failed gradually under compressive bending.

For the 2.5DSBWC beam, unexpectedly, the strain on the outer surface (i.e., points-1) increased only modestly until the specimen began to yield, as shown in Figure 7c. This was indicative of the high stress concentration in this region. However, the compressive strain on the inner surface (i.e., points-3) behaved quite interestingly. The strain at point-3 (failure region) remained approximately constant with increasing loading time of up to 8 min, but increased sharply thereafter until the beam failed. This attributed to the high stress concentration induced by the yarn waviness in this region. The strain at point-2 behaved similarly to that of point-3, indicating that the compressive stress was transferred from point-3 to point-2, which was a desirable failure mode for stiffeners.

For the 3DOWC beam, the tensile strain at the outer surface (i.e., points-1) remained largely unchanged but experienced a significant increase about 8 min into the test (see Figure 7d). Furthermore, the strains at point-3 and point-2 also remained unchanged till the beam failed. This unusual but interesting behavior of the strains indicated that its outer surface was subjected to tensile load, resulting in weft yarn stretching. Therefore, the stress concentration was induced by the combined effect of weft yarn stretching and Z-yarn inclination. More importantly, almost all loads were concentrated on the tension surfaces, which was borne by the weft fibers.

### 4.5. Micro X-ray CT Scan Analysis

While the previous section demonstrated the differences between the 2D, 2.5D, and 3D woven fabric L-beams, in terms of beam deformation when subject to bending, the effect of weaving structure on the failure mode of these curved components were essential to help composite part design and failure prediction.

#### 4.5.1. 2DPWC Beams

The X-ray CT scan results were shown in Figure 8. The porosity analysis of the scans, shown in Figure 8a, indicated that the cracks were formed in the support arms rather than in the center of the curved regions. This demonstrated that the stress concentration occurred mainly in the support arms, leading to the generation of cracks, as verified by the strain distribution of the specimen (see Figure 8a).

Crack propagation in the cross-section of the 2D beam along the a-a’ and b-b’ planes was shown in Figure 8b,c. The delamination followed a tortuous path occurred mainly between the first and third plies due to out-plane tensile stress (see Figure 8b), and did not extend across the thickness of the specimen, which was observable due to the low degree of nested layers. In the b-b’ plane, weft yarn breakage was caused by the compressive stress, observable on the inner-surface of the support arms, resulting in the crack propagation along the weft direction without yarn breakage, indicating that the failure was dominated by shear and delamination (see Figure 8c).

The above failure modes were consistent with the low tensile strength of the 2D beams, confirming that the 2D laminate failure was dominated by delamination, primarily due to the absence of Z-direction fibers.

#### 4.5.2. 2.5DSSWC Beams

The corresponding X-ray CT scan results of 2.5DSSWC were shown in Figure 9. The failure (i.e., the red region) occurring in 2.5DSSWC differed significantly from that in 2DPWC, as shown in Figure 9a, indicating that the initiation of the stress concentration was restricted by the woven structure.

In the a-a’ plane, the damage occurred in the regions containing crimped yarns, then propagated along the warp direction (i.e., X-direction) and extended to adjacent layers. This was attributed to the compressive circumferential stress in the regions of interlock yarns, and to the yarn-bridging, subsequently resulted in delamination and weft yarn breakage.

In the b-b’ plane, delamination appeared and propagated along the weft direction in the curved area where the compressive circumferential stress was the highest. The shear failure caused by the compressive stress was also observed. However, the cracks in this plane were shorter than those in the a-a’ plane, indicating that the crack propagation was intercepted by the crimped warp yarns. In this case, weft yarn breakage resulting from compression and shear also occurred on the inner surface.

The above failure analyses indicated that the failure of 2DPWC and 2.5DSSWC was primarily via delamination, combined with shear failure, induced by interlaminar shear and compressive stress, respectively. Then, the subsequent interlaminar crack was propagated along both the weft and warp directions, without any fiber breakage prior to shear failure. The results indicated that the interlaminar shear strength of 2DPWC and 2.5DSSWC depended on the yarn waviness and interlaminar shear strength between the plies. Hence, the 2.5DSSWC exhibited poor delamination resistance and the 2.5DSSW structure yielded no interlaminar strength improvement.

#### 4.5.3. 2.5DSBWC Beams

Similar to 2DPWC and 2.5DSSWC beams, the main damage area (i.e., the purple region) of 2.5DSBWC beam occurred in the curved region, but the failure spot was different (see Figure 10a). The damage areas in 2.5DSBWC beam was larger than that of 2.5DSSWC and their distribution was more scattered, indicating that the yarn crimp, which was severe in 2.5DSBWC, promoted stress concentrations under compressive bending. Moreover, the poor damage continuity with small voids (i.e., blue region) indicated that the crack formed was initiated by cracks either in resin crack or from delamination.

The failure mode of 2.5DSBWC was shown in Figure 10b,c. It was evident that delamination caused by the compressive circumferential stress was the dominant failure mode in the plies adjacent to the inner surface of the a-a’ plane. Due to the high yarn waviness, the delamination was restricted by significant yarn-bridging. In addition, matrix resin cracks occurred close to the neutral axis area due to the tensile circumferential stress resulting from fiber buckling.

In the b-b’ plane, the delamination area in 2.5DSBWC was smaller than that of 2.5DSSWC, due to its higher warp yarn waviness. Thus, compared with the energy required for the cracks to propagate along the other direction yarns, more energy was required for the crack initiation and propagation along the weft yarns. Eventually, weft-yarn would break as a result of shear failure, possibly due to the similarity between the 2.5DSBWC and 2.5DSSWC weaving styles.

Consequently, the 2.5DSBWC failed primarily via delamination and shear failure, together with weft yarn breakage. Compared to 2DPWC and 2.5DSSWC, 2.5DSBWC experienced a higher failure load as well as less damage and delamination, owing to the higher degree of warp yarn waviness. As a result, the 2.5DSBW structure was superior to the other structures, in terms of reinforcing the L-beams under compressive bending.

#### 4.5.4. 3DOWC Beams

The failure mode of the 3DOWC was significantly different from that of the other three L-beams. The damages regions in 3DOWC occurred quite extensively in both the curve area and the supporting arms (see Figure 11a).

The inter-ply delamination was concentrated in regions containing no Z-yarns, i.e., between the warp yarns and weft yarns, and the cracks propagated along the warp yarns (see Figure 11b,c). Moreover, the Z-yarns breakage caused by the tensile circumferential stress in the a-a’ plane indicated that crack propagation was restricted by these fibers. Therefore, crack propagation was forced to go along higher energy routes along the warp/X-direction, leading to the higher load-bearing capacity of the 3DOWC.

However, in the b-b’ and b_1_-b_2_ planes, delamination appeared mainly in the first outer warp ply of the beam, propagated along the Z-yarns, which was different from the other L-beams. The corresponding stress concentration (i.e., tensile circumferential stress) was, accordingly, generated primarily in this region.

In addition, the failure occurring in the Z-yarn-containing region was noticeably different from the failure occurring in the Z-yarn-free region (see Figure 11d,e). In the Z-yarn-free regions, the delamination propagated easily and extensively along the weft yarns, but the cracks were discontinuous in regions along the Z-direction, where the weft yarns restricted the crack propagation.

The failure modes occurring in the Z-yarn-containing regions, on the other hand, were different. For example, delamination in the outer surface region was not continuous, because the Z-yarns blocked the crack propagation, but could not prevent new cracks from forming. Moreover, continuous crack propagations along the Z-yarns were observed in the through-thickness direction. This was attributed to the absence of weft yarns, which made it easy for the stress to be transferred along the Z-yarns, causing the delamination along the warp yarns and Z-yarns.

The above failure mode analyses illustrated that the failure of the 3DOWC beam under compressive bending was dominated by delamination, followed by Z-yarn breakage. As a result, although the 3DOWC exhibited higher load-carrying capacity than the other three woven styles (i.e., 2DPW, 2.5DSSW, and 2.5DSBW), it did not necessarily provide better delamination resistance compared to the 2.5D composites whose yarns were highly crimped.

## 5. Conclusions

This study demonstrated a viable option of using highly crimped 2.5D woven fabrics to fabricate composite stiffeners having compressive bending loads and interlaminar shear resistance equivalent to or better than that of the expensive 3D composite approach. L-beams were selected as representative stiffener components. L-beams of two types of 2.5D composites, namely 2.5D shallow-straight and shallow-bend woven fabric composites, together with a 2D plain weave and 3D woven fabric reference composites, were prepared from a standard modulus carbon fibers and subsequently infused with epoxy resins. The bending strength, deformation, and failure mode of these L-beams were obtained using a custom-built bending test fixture, and the data were acquired using DIC and Micro X-ray CT scans.

The study further revealed that the composite fabrication process, namely VARI, had a significant effect on the bending strengths of the L-beams, particularly at the through-thickness direction, by altering the fiber orientation and/or yarn waviness. Such alterations generally did not exist in 2D composite fabrications, and should be considered when designing 2.5D and 3D composite parts. While the presence of Z-yarns were favorable in term of achieving high through-thickness performances, similar effects could be obtained via the utilization of highly crimped warp yarns, such as in the case of 2.5DSSWC.

As factors controlling the stability of the stiffeners, the L-beams’ stiffness and strain distribution were controlled by their woven structures. The data in this work demonstrated that the conventional 2D laminate stiffeners that had their maximum strains in the supporting arms, therefore had low stiffness than the 2.5D and 3D L-beams, as expected.

The outcome of this study would provide guidance to the selection of appropriate stiffeners, as well as help designing higher performance composite structures, both effectively and economically. The data thus collected would enrich the database of 2.5D and 3D woven composites for further numerical simulation via finite element modeling.

## Figures and Tables

**Figure 1 materials-12-02221-f001:**
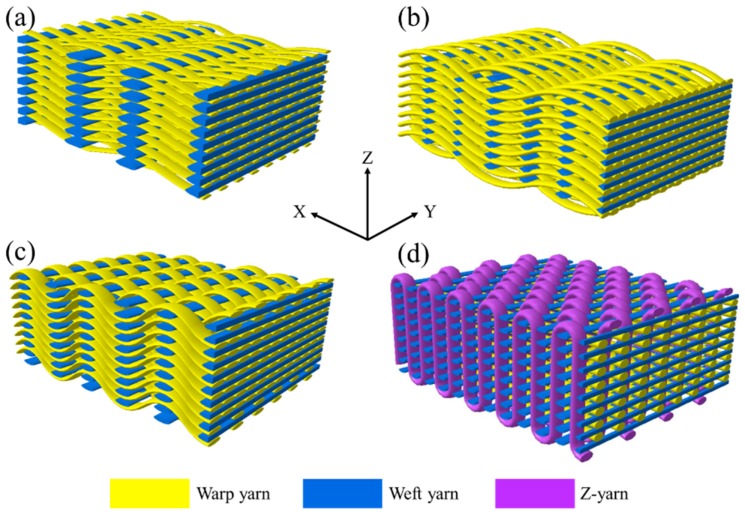
Illustration of woven fabric structures. (**a**) 2DPW; (**b**) 2.5DSSW; (**c**) 2.5DSBW; and (**d**) 3DOW.

**Figure 2 materials-12-02221-f002:**
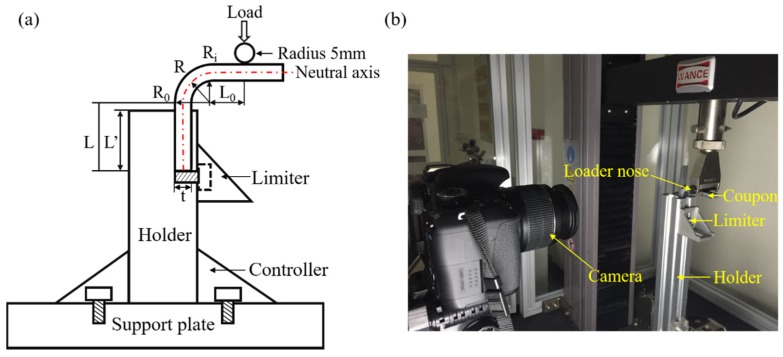
Compressive bending test design. (**a**) schematic demonstration of bending test preformed on L-beams; (**b**) quasi-static compressive bending test of L-beams.

**Figure 3 materials-12-02221-f003:**
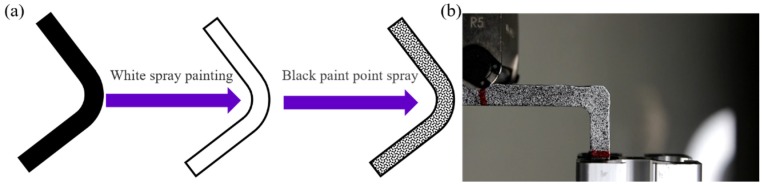
Digital Image Correlation (DIC) test coupons. (**a**) Test coupon preparation illustration; (**b**) actual test specimen for optical deformation measurement.

**Figure 4 materials-12-02221-f004:**
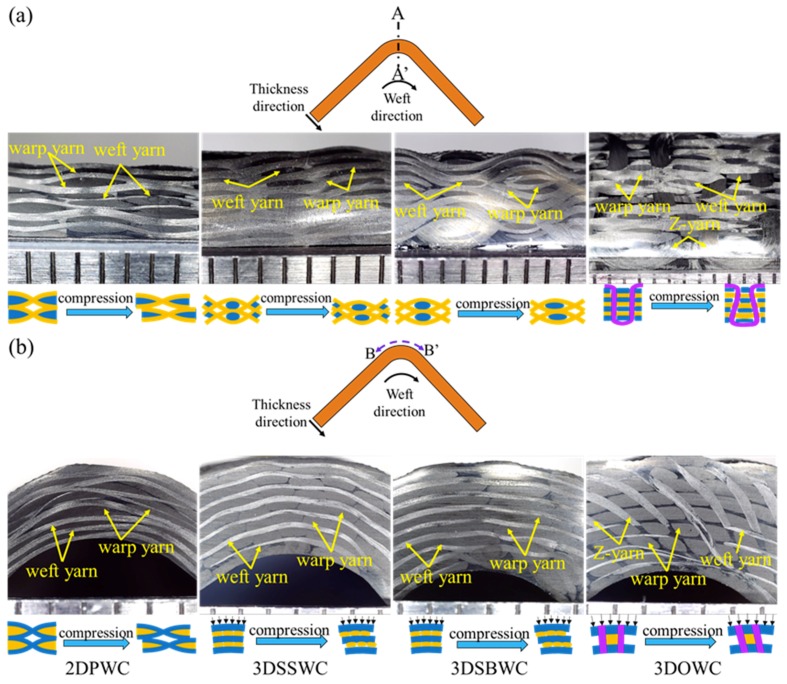
Cross-section of L-beams along the (**a**) A-A’ plane and (**b**) B-B’ plane.

**Figure 5 materials-12-02221-f005:**
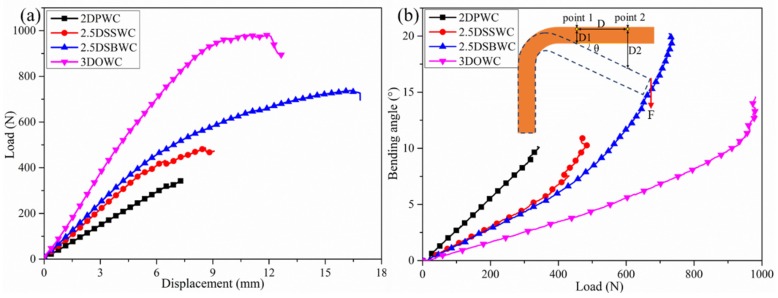
Compressive bending of L-beams. (**a**) Load-displacement curves; (**b**) bending angle-load curves.

**Figure 6 materials-12-02221-f006:**
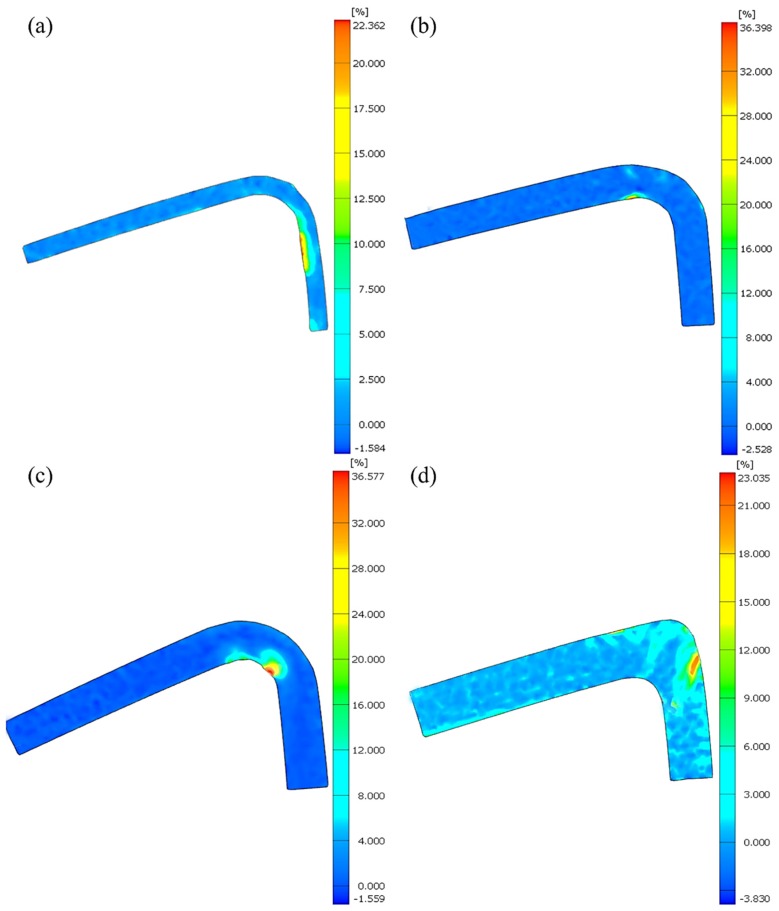
Strain field distributions of L-beams at the failure. (**a**) 2DPWC, (**b**) 2.5DSSWC, (**c**) 2.5DSBWC, and (**d**) 3DOWC.

**Figure 7 materials-12-02221-f007:**
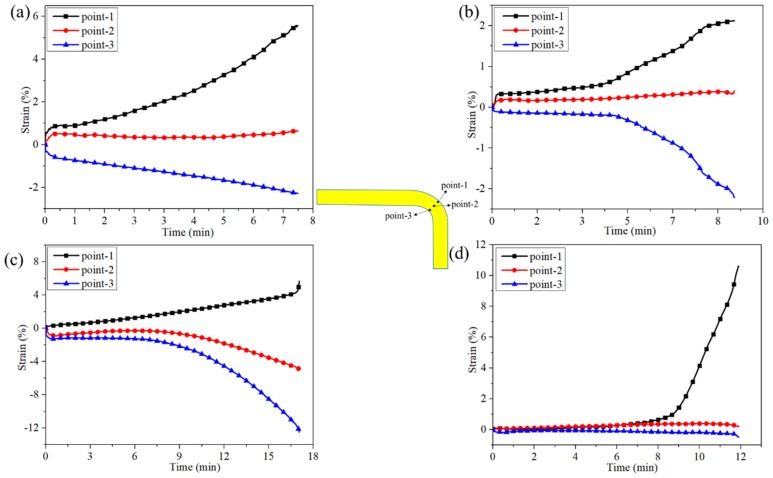
Strain field distributions of 3D composites during the compressive bending test. (**a**) 2DPWC, (**b**) 2.5DSSWC, (**c**) 2.5DSBWC, and (**d**) 3DOWC.

**Figure 8 materials-12-02221-f008:**
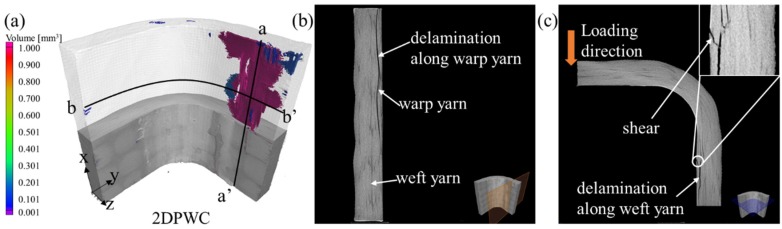
X-ray CT results of the 2D L-beams. (**a**) void analysis; (**b**) damage along the a-a’ direction, and (**c**) damage along b-b’ direction.

**Figure 9 materials-12-02221-f009:**
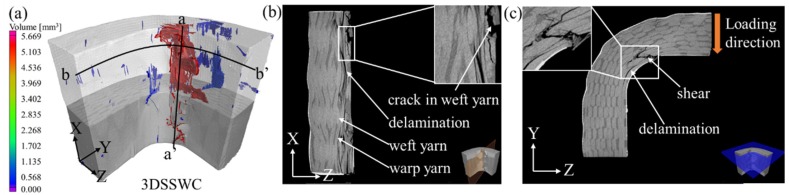
X-ray CT results of the 2.5DSSWC L-beams. (**a**) void analysis; damage along the (**b**) a-a’ direction; and (**c**) damage along b-b’ direction.

**Figure 10 materials-12-02221-f010:**
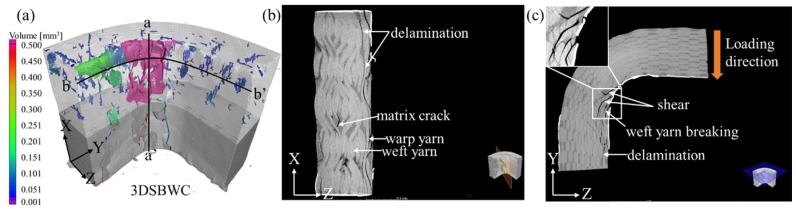
X-ray CT scans of the 2.5DSBWC L-beam. (**a**) void analysis; damage along the (**b**) a-a’ direction; (**c**) damage along b-b’ direction.

**Figure 11 materials-12-02221-f011:**
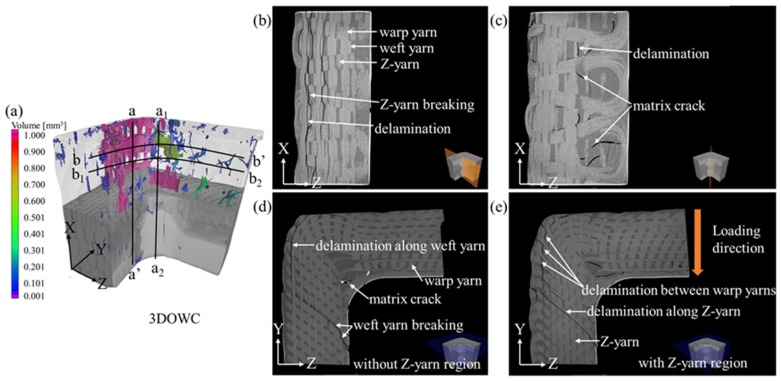
X-ray CT scans of the 3DOWC L- beam. (**a**) void analysis; damage along the (**b**) a-a’ direction; (**c**) damage along a_1_-a_2_ direction; (**d**) damage along b-b’ direction; (**e**) damage along b_1_-b_2_ direction.

**Table 1 materials-12-02221-t001:** The properties of SYT55S-12K carbon fiber.

Type	Yarn Density g/km	Tensile Strength (MPa)	Tensile Modulus (GPa)	Elongation at Break (%)	Sizing Content (%)
**SYT55S-12K**	447–452	5890–6222	290–292	2.0–2.1	1.35–1.43

**Table 2 materials-12-02221-t002:** The properties of epoxy resin BAC 172.

Type	Density (g/cm^3^)	Tensile Strength (MPa)	Tensile Modulus (MPa)	Tensile Elongation %	Flexural Strength (MPa)	Flexural Modulus (MPa)
**BAC 172**	1.2	75	2850	7.6	130	3500

**Table 3 materials-12-02221-t003:** Weaving parameters of the fabrics.

	2DPW	2.5DSSW	2.5DSBW	3DOW
**Linear density (tex)**	450			
**Yarn density (ends/cm)**	warp:3.4	warp: 9	warp: 9	warp(z): 9
	weft:3.2	weft: 3	weft: 3	weft: 3
**Thickness (mm)**	2.72	5.4	5.2	7
**Layers (warp×weft)**	8 × 8	8 × 8	8 × 9	8 × 9

**Table 4 materials-12-02221-t004:** L-beam and fixture dimensions.

Structures	L (mm)	L’ (mm)	L_0_ (mm)	R_0_ (mm)	w (mm)	t (mm)
**2DPW**	45.1	34.5	30	6.2	14.8	2.2
**2.5DSSW**	45.1	34.5	30	8.1	14.8	4.3
**2.5DSBW**	45.1	34.5	30	8.6	14.6	4.9
**3DOW**	44.5	34.5	30	9.9	14.7	6.8

**Table 5 materials-12-02221-t005:** Main factors affecting the compaction of woven fabrics [27,28,29,30,31,32].

Factors	Uncompressed	Compressed
**Void/gap Accumulation and Compression**	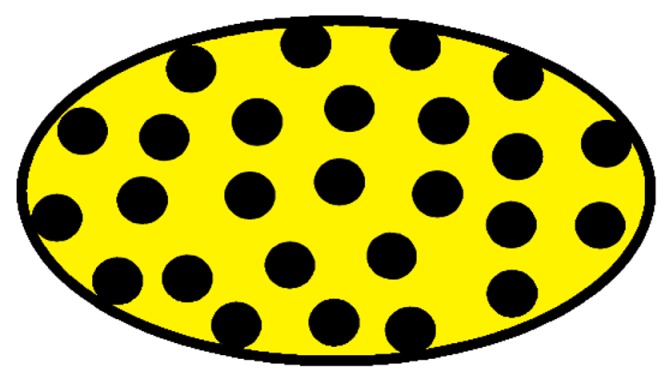	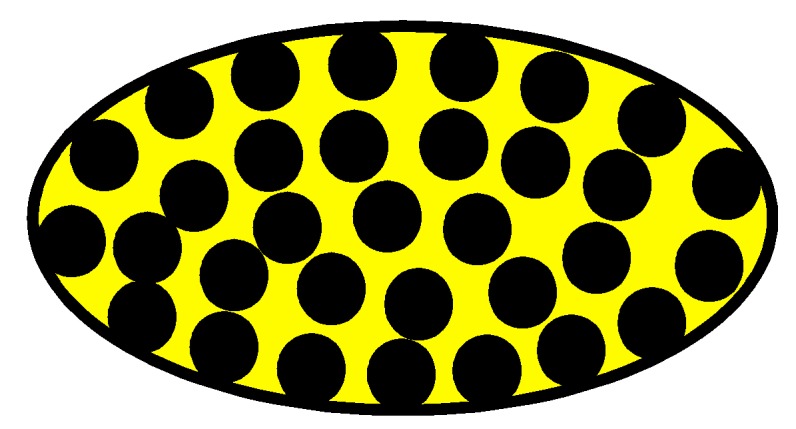
**Yarn Cross-Section Deformation**	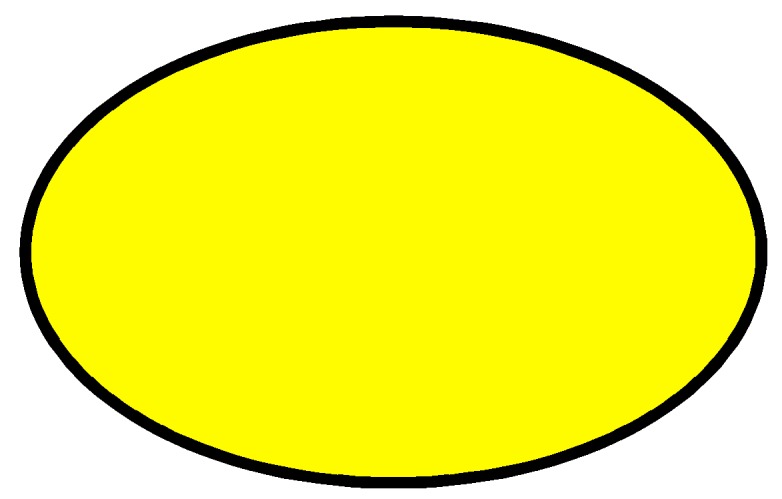	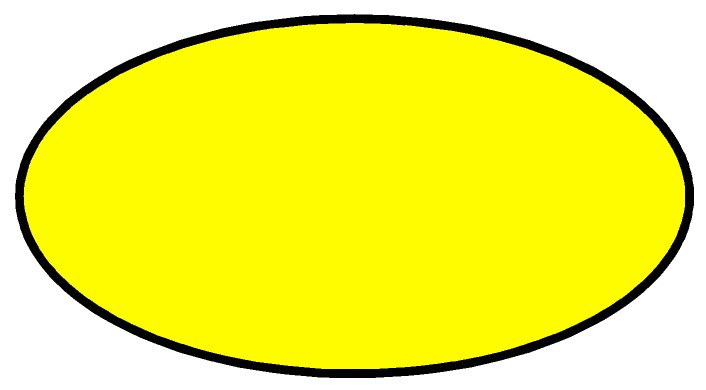
**Yarn Flattening**	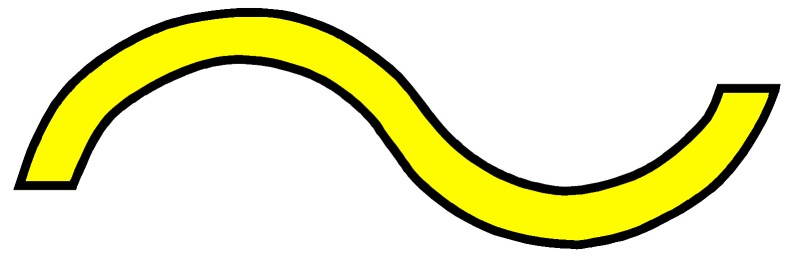	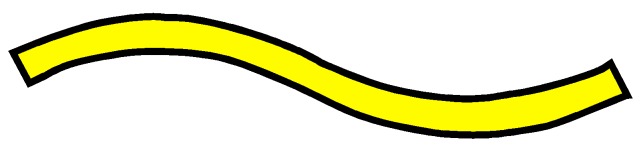
**Yarn Bending Deformation**	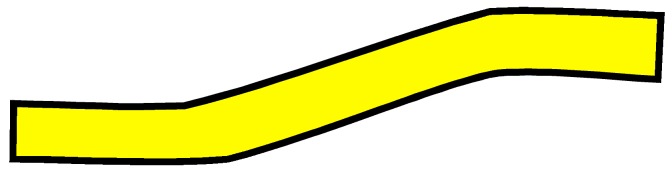	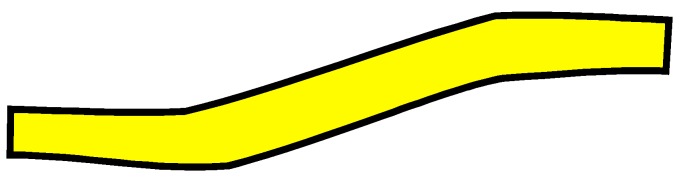
**Nesting and Slipping**	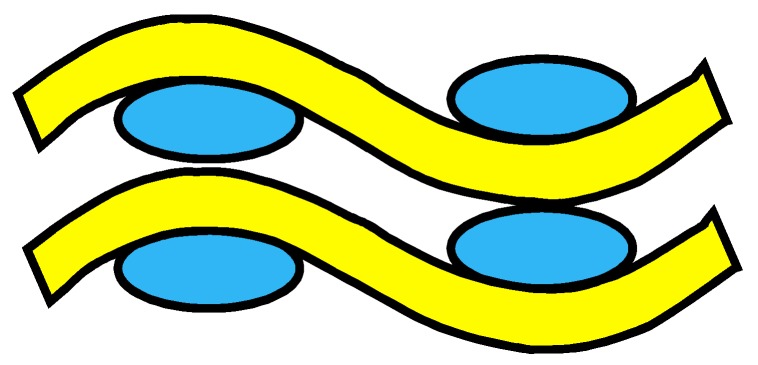	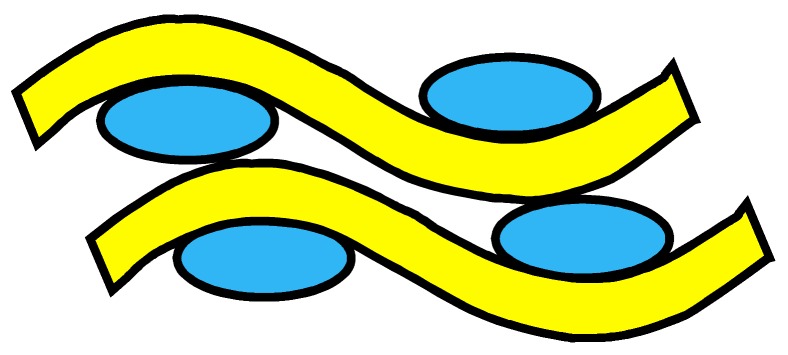

**Table 6 materials-12-02221-t006:** Tensile and compressive strengths of L-beams.

Structure	Bending Load F/N	Compressive Stress σ_c_/MPa	Tensile Load * F/N	Tensile Stress * σ_t_/MPa
2DPWC	256	7.8	156	3.4
2.5DSSWC	487	7.7	285	3.5
2.5DSBWC	738	10.3	3221	37.9
3DOWC	984	9.8	1885	15.2

* Data cited from earlier publication [8].

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
