# Peer review of "The Failure Mechanism of Composite Stiffener Components Reinforced with 3D Woven Fabrics"

_materials, 2019, doi:10.3390/ma12142221_

Reviewer 1 Report

1. The originality and the scientific value of the subject are good. An important problem having direct application is treated.

2. The Abstract of the manuscript is concrete and gives a good summary of this work.

However, given that the overall text contains many abbreviations, it is the reviewer’s opinion that the authors should definitely add a nomenclature at the beginning of the manuscript.

 3. The Introduction Section in its current form is not well written and adequate.

In this context, I strongly recommend the authors to further analyze and discuss the results of Refs. [1-4], [7-12] and [13-17]. In addition, the differences/advantages of the present investigation compared to other literature works should be written out at the end of this Section in a much more thorough and comprehensive manner.

4. The materials, their implementations, applied methods and especially the use of the investigated material are explained in detail. The composition, the origin of the material used, dimensions of specimens etc are all mentioned.

5. Presentation of the experimental work is very thorough. Process and prerequisites of sample preparation are clearly mentioned. However, the authors are kindly recommended to provide some further technical details about the laboratory equipment that they used to carry out their experiments.

 6. The presentation and clarity of results and data are good.

However, the discussion of the results is relatively adequate. In this context, the authors could give some additional theoretical explanations about Figs. 3, 4 and 6. Moreover, is there any possibility for comparison with theoretical and/or advanced computational methods (FEM, BEM)?

7. Logic and coherence are concrete and the clarity and quality of writing are sound.

 8. The Conclusions Section performs the findings of this work in an adequate manner.

However, I invite the authors to add a paragraph on the motives and prospects that this work provides for future research.

Overall, it can be said that the manuscript may be recommended for publication provided that the authors interpret these critical remarks in a constructive manner and revise the manuscript accordingly. 

Author Response

Point 1: The originality and the scientific value of the subject are good. An important problem having direct application is treated.

 Response 1: Thank you.

 Point 2: The Abstract of the manuscript is concrete and gives a good summary of this work. However, given that the overall text contains many abbreviations, it is the reviewer’s opinion that the authors should definitely add a nomenclature at the beginning of the manuscript.

 Response 2: The nomenclature was added at the beginning of the manuscript. (see Line 39)

 Point 3: The Introduction Section in its current form is not well written and adequate.

In this context, I strongly recommend the authors to further analyze and discuss the results of Refs. [1-4], [7-12] and [13-17]. In addition, the differences/advantages of the present investigation compared to other literature works should be written out at the end of this Section in a much more thorough and comprehensive manner.

 Response 3: Agreed. Refs. [1-4], [7-12] and [13-17] was further analyzed and discussed in a few additional sentences, and the differences/advantages of the present investigation compared to other literature works was added. (See Line 43, 50-63,64-75)

 Point 4: The materials, their implementations, applied methods and especially the use of the investigated material are explained in detail. The composition, the origin of the material used, dimensions of specimens etc are all mentioned.

 Response 4: Thank you.

 Point 5: Presentation of the experimental work is very thorough. Process and prerequisites of sample preparation are clearly mentioned. However, the authors are kindly recommended to provide some further technical details about the laboratory equipment that they used to carry out their experiments.

 Response 5: The technical details about the laboratory equipment used to carry out the experiments were added. (See section 2.3)

 Point 6: The presentation and clarity of results and data are good. However, the discussion of the results is relatively adequate. In this context, the authors could give some additional theoretical explanations about Figs. 3, 4 and 6. Moreover, is there any possibility for comparison with theoretical and/or advanced computational methods (FEM, BEM)?

 Response 6: The stress analysis of the L-beams was added to further explain the original Fig. 3 (Fig. 5 in the current revised manuscript). We do not have reliable ways to conduct strain analysis for Figures 4 and 6, either via FEM or BEM. However, we will take your suggestions and conduct these analyses in the future.

 Point 7: Logic and coherence are concrete and the clarity and quality of writing are sound.

 Response 7: Thank you.

 Point 8: The Conclusions Section performs the findings of this work in an adequate manner. However, I invite the authors to add a paragraph on the motives and prospects that this work provides for future research.

 Response 8: The paragraph on the motives and prospects that this work provides for future research was added. (See section 5)

Reviewer 2 Report

1.       In Fig. 4: the strain map scale font is hardly readable and must be enlarged.

2.       Use of the acoustic emission technique (cf. Kubiak et al 2015) could help in defect type recognition while loading.

3.       Did the authors consider in their own studies the effect of different radii of the L-shape corner on delamination initiation? Were the specimens 100% free from defects while inspected before loading? (cf. the papers by Teter et al.)

4.       In the paper several different kinds of defects are indicated. In the future the Authors should define them more precisely. For example, when delamination goes through the fabric layer the bridging phenomenon and the fracture process zone must be considered. Also the distinction must be made among different fracture modes (pure or mixed – see the recent papers by Samborski et al).

Author Response

Point 1: In Fig. 4: the strain map scale font is hardly readable and must be enlarged.

 Response 1: Agreed. The strain map scale font has been enlarged. (See Fig. 6)

 Point 2: Use of the acoustic emission technique (cf. Kubiak et al 2015) could help in defect type recognition while loading.

 Response 2: Agreed. Since we do not have access to an acoustic instrument, the DIC test and X-ray CT scan were used instead.

 Point 3: Did the authors consider in their own studies the effect of different radii of the L-shape corner on delamination initiation? Were the specimens 100% free from defects while inspected before loading? (cf. the papers by Teter et al.)

 Response 3: Due to the high cost and difficulty of weaving the 3D stiffeners having different radii, we selected the most common construction of the L-shape corner on delamination was considered. Furthermore, the objective of this research is to determine the properties of stiffeners having different woven structures, not to get into details of the effectives of different radii. Therefore, the inner radius of each specimen was kept constant (4mm).

 The specimen cross-section was inspected by metallographic microscope and no porosity was observed.

 Point 4: In the paper several different kinds of defects are indicated. In the future the Authors should define them more precisely. For example, when delamination goes through the fabric layer the bridging phenomenon and the fracture process zone must be considered. Also the distinction must be made among different fracture modes (pure or mixed – see the recent papers by Samborski et al)

 Response 4: Agreed. The defect types were defined according to the different stress they suffered and added to the revision.

Reviewer 3 Report

Dear Authors,

the paper has several flaws, and there are some weaknesses through the manuscript which need improvement. The submitted manuscript cannot be accepted for publication in this form, but it has a chance of acceptance after revise and resubmit. My suggestions and comments are as follows:

1-         Abstract cover the results, but give a little information about the purpose and the procedure of the tests. It is just mentioned that bending test is conducted. I think the abstract must be rewrite.

2-         The authors do not present a clear justification of why their work is important. Motivation of the work should be added to introduction.

3-         Several papers are cited in the manuscript, but they are not reviewed accurately. For instance, Ref. [7-12] in introduction. Each reference should be commented at least in a couple of sentence. How the previous studies tied to the current manuscript?

4-         The literature survey must be enhanced at least by considering and citing all the following relevant works:

a.         Composites Part B: Engineering, 160:306-314 (2019)

d.         Materialwissenschaft und Werkstofftechnik, 48:753-759 (2017)

5-         It is necessary to remove the last paragraph in introduction “This section may be divided by subheadings. It should provide a concise …” It seems this paragraph is remaining from the template.

6-         The title “Experimental” must be modified, it is not completed. The tables (S1 to S5) are presented as supplementary materials. However, I think this data should be presented in the main text of the manuscript (present as supplementary data is not necessary).

7-         As composite manufacturing process is presented in the manuscript, it is recommend to read and cite following published papers:

a.         Composites Part A: Applied Science and Manufacturing, 120:127-139 (2019)

b.         International Journal of Mechanical Sciences, 141:223-235 (2018)

8-         Since Fig. 1 and Fig. 2 are already published in Ref. [6], authors must refer to this paper or remove/change the figures to reduce similarity (same issue for the Fig. S1 and Fig. S4). Also, full pagination of Ref. [6] is needed in the reference list. 

9-         Please explain advantages and disadvantages of the strain gages. Why authors didn’t used strain gauges? What is the error in the measurement with applied DIC technique?

10-       How the standard deviation is calculated? Details of DIC analysis must be presented. The legend in Fig. 4 is totally unreadable (same issue for the legend in Fig. 6 to Fig. 9). 

11-       It looks better if you provide numerical calculations of bending test and the strength of the examined L-shape composite beams. Indeed, presenting all experimental results looks good, but determining strength according to the formula is needed, so it must be added. 

12-       In its language layer, the manuscript should be considered for English language editing, and the general sentences should be removed from the manuscript.

13-       The conclusion must be more than just a summary of the manuscript, and it should be more accomplish the goals, and I think the conclusion must be rewrite. Moreover, the reference list must be updated and completed.

 Author Response

The paper has several flaws, and there are some weaknesses through the manuscript which need improvement. The submitted manuscript cannot be accepted for publication in this form, but it has a chance of acceptance after revise and resubmit. My suggestions and comments are as follows:

Point 1: Abstract cover the results, but give a little information about the purpose and the procedure of the tests. It is just mentioned that bending test is conducted. I think the abstract must be rewrite.

 Response 1: Agreed. The abstract was rewritten.

 Point 2: The authors do not present a clear justification of why their work is important. Motivation of the work should be added to introduction.

 Response 2: Agreed. The motivation of the work has been added to introduction. (See Line 64-75)

 Point 3: Several papers are cited in the manuscript, but they are not reviewed accurately. For instance, Ref. [7-12] in introduction. Each reference should be commented at least in a couple of sentences. How the previous studies tied to the current manuscript?

 Response 3: Agreed. Refs. [1-4], [7-12] and [13-17] was further analyzed and discussed in a few more sentences. (See Line 43, 50-63,64-75)

 Point 4: The literature survey must be enhanced at least by considering and citing all the following relevant works:

a.         Composites Part B: Engineering, 160:306-314 (2019)

d.         Materialwissenschaft und Werkstofftechnik, 48:753-759 (2017)

 Response 4: Thank you for your suggestion, the literature survey has been enhanced and the suggested reference has been cited. (See Line 43)

 Point 5: It is necessary to remove the last paragraph in introduction “This section may be divided by subheadings. It should provide a concise …” It seems this paragraph is remaining from the template.

 Response 5: The sentence “This section may be divided by subheadings. It should provide a concise …” in the introduction has been deleted.

 Point 6: The title “Experimental” must be modified, it is not completed. The tables (S1 to S5) are presented as supplementary materials. However, I think this data should be presented in the main text of the manuscript (present as supplementary data is not necessary).

 Response 6: Thank you for your suggestion. The title “Experimental” has been changed into “Materials and experimental methods”. The supplementary data has been presented in the main text of the manuscript. (See Table 1 to 4 in the revised manuscript)

 Point 7: As composite manufacturing process is presented in the manuscript, it is recommend to read and cite following published papers:

a.     Composites Part A: Applied Science and Manufacturing, 120:127-139 (2019)

b.     International Journal of Mechanical Sciences, 141:223-235 (2018)

 Response 7: Thank you for your suggestion, the references suggested was cited in the manuscript. (See Line 210, 214)

 Point 8: Since Fig. 1 and Fig. 2 are already published in Ref. [6], authors must refer to this paper or remove/change the figures to reduce similarity (same issue for the Fig. S1 and Fig. S4). Also, full pagination of Ref. [6] is needed in the reference list.

 Response 8: Agreed. The original Fig.1 and Fig.2 have been modified. (See the Fig. 1 and Fig. 4 in the revised manuscript) Fig.S1 and Fig. S4 have been removed and reference [6] was cited. The full pagination of Ref. [6] was added in the reference list.

 Point 9: Please explain advantages and disadvantages of the strain gages. Why authors didn’t used strain gauges? What is the error in the measurement with applied DIC technique?

 Response 9: The strain gages are normally used in flat panels. It is not suitable for coupons subjected to complex load (i.e. containing both tension and compression) or curved shapes. Therefore, DIC was used to measure the strain distribution, because it has the capacity to measure full-field strains instead of strains in a limited area.

 The error in the measurement with applied DIC technique was related to the camera resolution/pixels, according to the study of Jason Ingham (Composite Structures, 212(15), 2019, 43-57). The strain difference was around 0.06%.

 Point 10: How the standard deviation is calculated? Details of DIC analysis must be presented. The legend in Fig. 4 is totally unreadable (same issue for the legend in Fig. 6 to Fig. 9).

 Response 10: The standard deviation for the strains was calculated by comparing extensometer with strain gauges according to reference [26], our tests were conducted according to Jason Ingham (Composite Structures, 212(15), 2019, 43-57), therefore we believe the deviations of our measurements are the same. Details of DIC analysis were also added (See section 3.2). The legends in the original Fig. 4 and Fig.6-Fig.9 (see Fig. 6 and Fig. 8-11in this revised manuscript) were enlarged.

 Point 11: It looks better if you provide numerical calculations of bending test and the strength of the examined L-shape composite beams. Indeed, presenting all experimental results looks good, but determining strength according to the formula is needed, so it must be added.

 Response 11: Agreed. The numerical calculations of the bending test and the strength of the examined L-shape composite beam were added. (See section 4.3)

 Point 12: In its language layer, the manuscript should be considered for English language editing, and the general sentences should be removed from the manuscript.

 Response 12: The manuscript has been re-examined and edited.

 Point 13: The conclusion must be more than just a summary of the manuscript, and it should be more accomplish the goals, and I think the conclusion must be rewrite. Moreover, the reference list must be updated and completed.

 Response 13: The conclusion has been rewritten and the reference was updated.

 Round  2

Reviewer 1 Report

The authors responded to my critical remarks and made adequate improvements to their article.

In this framework, I am satisfied with the manuscript in its current form and I recommend it for publication.